# Hybrid Centralized-Decentralized Economic Dispatch Based on A Distributed Finite-Step Consensus Algorithm with Divided Regional Incremental Costs

1st Hanwen Zhang
the School of Automation Engineering
University of Electronic Science and Technology of China
Chengdu, China
hwzpiaopiao@126.com

2nd Hanqing Yang
the School of Automation Engineering
University of Electronic Science and Technology of China
Chengdu, China
hqyang5517@uestc.edu.cn

4th Tieshan Li
the School of Automation Engineering
University of Electronic Science and Technology of China
Chengdu, China
tieshanli@126.com

3rd Yue Long
the School of Automation Engineering
University of Electronic Science and Technology of China
Chengdu, China
longyue@uestc.edu.cn

*Abstract*—The economic dispatch problem of power systems, which is typically based on multi-agent networks, often necessitates conditions like full graph connectivity. However, as the scale of power systems expands, maintaining full graph connectivity in communication systems could result in increased costs and communication burdens. Consequently, this paper investigates a distributed finite-step consensus method that incorporates geographical region division, aiming to achieve economical power allocation without the need for full graph connectivity.

In this method, we initially derive a Laplace matrix from the distributed topology of generators in multi-agent power systems. Subsequently, we construct another Laplace matrix by geographically dividing regions, thereby illustrating the communication relationships between these regions. Subsequently, we calculate the incremental cost of each region, taking into account the power constraints of the generators. Moreover, to expedite convergence, we employ a distributed finite-step consensus algorithm to address the economic dispatch problem, leveraging the divided regional incremental costs. Finally, we validate the effectiveness and accuracy of the proposed method through results on various multi-agent topologies and comparisons with other iterative algorithms.

*Index Terms*—Hybrid centralized-decentralized economic dispatch, distributed algorithm, multi-agent network system, finite-step consensus algorithm.

## I. INTRODUCTION

In recent times, economic dispatch (ED) has assumed significant importance in maintaining power balance and optimizing operations in modern power systems. A well-implemented ED can not only address issues of power shortage or surplus but also enhance economic profitability[1], [2], [3]. Most existing studies primarily focus on generator costs, with an aim to minimize these costs while maximizing profits. To date, numerous methods have been developed to tackle ED problems, including heuristic methods[4], [5], [6] and reinforcement learning-based methods[7], [8], [9].

The aforementioned scheduling policies are implemented in a centralized fashion, where the control center plays a crucial role[10]. A genetic algorithm was proposed by [11], which adopted new crossover and mutation operations for economic scheduling based on valve point loads. [12] proposed a mathematical model-based analysis strategy that solved economic, emission, and joint economic and emission scheduling problems through a single equivalent objective function. In [13], a small unit with only three generators and one energy storage battery was designed, using a centralized control method for smart grid ED that does not consider network equality and security constraints. If the system's generator units cannot meet the power demand, storage units will be dispatched. [14] utilized a differential evolution algorithm to address the centralized control ED problem associated with non-smooth cost functions. However, due to advancements in science and technology, the traditional centralized control strategy has grown increasingly inadequate for modern power systems, given its heavy reliance on the control center. The proliferation of distributed generators in modern power systems results in escalated communication system costs when centralized control schemes are implemented. Worse still, the efficiency will diminish in the event of a central controller failure.

Presently, the control strategies of multi-agent systems are progressively evolving towards diversification. Distributed control strategies, particularly those based on multi-agent distributed algorithm optimization, have garnered increasing attention due to their flexibility and efficiency. In [15], a new consensus based economic scheduling algorithm was

proposed and the impact of time delay on distributed economic scheduling was rigorously analyzed. Being fully distributed, this algorithm can facilitate optimal scheduling of energy resources in micro-grids in a decentralized fashion. Furthermore, distributed control is currently primarily employed to address economic scheduling issues[16]. [17] utilized frequency analysis methods to establish a critical upper bound for the delay of the ED algorithm and introduced a collaborative distributed control scheme for optimal active power control of multiple generators in micro-grids. [18] proposed a micro-grid distributed economic dispatching strategy that leverages multiple energy storage systems, thereby overcoming the challenge of dynamic coupling between all decision variables and random variables inherent in the centralized dispatching formula. A distributed controller achieved global control solely through information communication with directed interconnected neighbors, obviating the need for a control center [19]. By incorporating distributed methods into multi-agent systems, the ED problem can be resolved across various network topologies [20]. However, particularly in the context of large-scale, multi-agent-based power systems, power exchange typically occurs between geographically distinct regions. Consequently, the fully distributed method could impose a substantial communication burden, escalate computational complexity, and diminish power exchange efficiency.

Additionally, the convergence speed of the optimal algorithm is of paramount importance in practical applications. If the algorithm's convergence time is excessively long, it may hinder the practical application of the results before the algorithm converges. Owing to the distributed finite-step consensus algorithm, the optimal algorithm can converge within a shorter iteration time, thereby saving computational time and enhancing efficiency. [21] amalgamated the Newton-Raphson method, graph discovery algorithm, and finite time consistency algorithm to propose a novel distributed finite-step iterative algorithm for solving the economic scheduling problem in modern network physical power systems. This offered some valuable insights for the field of distributed fast computing and computation. In [22], a distributed algorithm based on a finite step consistency algorithm was proposed, which set the incremental cost of each distributed generation unit as a consistency variable, and all units only obtained the optimal value by exchanging information with neighbors. Furthermore, this algorithm converged to the optimal solution in a finite number of steps, thereby markedly enhancing efficiency.

Therefore, based on the above, a distributed finite-step consensus algorithm with regional division is proposed to solve the ED problem without a fully distributed communication system and improve the convergence speed. The contributions of this paper are summarized as follows:

1) In contrast to [23], the proposed method facilitates the economic operation of large-scale power systems by leveraging geographically divided regional incremental costs, thereby reducing communication costs and burdens without necessitating a fully distributed communication system.

2) This suggested proposed method considers the power constraint of each generator, which ensures that in practical application, the generators work within the rated power and will not appear in the overload.

3) The proposed algorithm in this paper will converge within a limited number of iteration steps and will achieve incremental cost consistency, which is more beneficial for practical applications with improving convergence speed.

## II. PRELIMINARY KNOWLEDGE

### A. Graph Theory

In this article, the multi-agent network, which is composed of three elements, is described by the weighted undirected graph $\mathcal{G} = (\mathcal{V}, \mathcal{E}, \mathcal{A})$, to describe the multi-agent network, which consists of three parts: 1) a vertex set $\mathcal{V} = \{v_1, v_2, \ldots, v_N\}$ to represent $N$ vertices in the graph; 2) an undirected edge set $\mathcal{E} \subseteq \mathcal{V} \times \mathcal{V}$ to describe all lines between vertices; and 3) an adjacency matrix $\mathcal{A} = \{a_{ij}\}_{N \times N}$ with a non-negative element $a_{ij} > 0$ if $(v_i, v_j) \in \mathcal{E}$ and $a_{ij} = 0$ otherwise.

The pair of nodes $(v_i, v_j)$ indicates an edge $e_{ij}$, which indicates that nodes $v_i$ and $v_j$ can communicate with one another. We take it for granted in this article that $a_{ij} = 0$ denotes the absence of a self-loop in the graph. The degree of node $v_i$ in the undirected graph is commonly defined as

$$d_i = \sum_{j=1, j \neq i}^{N} a_{ij}.$$

The Laplacian matrix, which is described as follows, is crucial to the study of the properties of a graph in graph theory.

$$\mathcal{L} = \mathcal{D} - \mathcal{A} \tag{1}$$

where $\mathcal{D} = \text{diag}\{d_1, d_2, \ldots, d_N\}$. A path between nodes $v_i$ and $v_j$ in the graph is a sequence of edges $(v_1, v_{i1}), (v_{i1}, v_{i2}), \ldots, (v_{ik}, v_j)$ with distinct nodes $v_{il} \in \mathcal{V}$. In addition, an undirected graph $\mathcal{G}$ is connected if and only if $\mathcal{L}$ is positive semi-definite and has a simple zero eigenvalue with $\mathcal{L}1_n = 0$, where $1_n = (1, 1, \ldots, 1)^\top \in \mathbb{R}^n$. For the Laplacian matrix $\mathcal{L}$, the upper bound of the maximum eigenvalue of $\mathcal{L}$ is often estimated to analyze the graph's properties, $\delta_1, \delta_2, \ldots, \delta_n$ are the characteristic value of $\mathcal{L}$.

### B. Problem Formulation

A multi-agent electric power system is used to collaboratively determine the best solutions to the considered ED problem. In a power grid, the cost function of power generation is typically defined as

$$f_i(P_i) = \alpha_i P_i^2 + \beta_i P_i + \gamma_i \tag{2}$$

where $\alpha_i$, $\beta_i$, and $\gamma_i$ are the parameters of the $i$th generating power, and $P_i$ is the $i$th generator power. We take the electricity grid to have a total of $N$ generators. The ED in the power system is to minimize the overall cost of the power grid with certain limits after accounting for the entire power demand and

the capacity of the power generation. This problem is related to the following:

$$\min_{P_i} \sum_{i=1}^{N} f_i(P_i)$$
$$\text{s.t.} \sum_{i=1}^{N} P_i = P_D \tag{3}$$
$$P_{\min} \leq P_i \leq P_{\max}$$

where $P_{\min}$ and $P_{\max}$ denote the lowest and maximum capabilities of the $i$th power generation, and $P_D$ represents the total power demand of the load in the power grid.

For every generator $\lambda_i$, the incremental cost is defined as

$$\lambda_i = \frac{\partial f_i(P_i)}{\partial P_i} = 2\alpha_i P_i + \beta_i \tag{4}$$

## III. PROPOSED ALGORITHM

We present a finite-step consensus Algorithm in this section to solve distributed multi-region partitioning using prior economic dispatch. First, the topic of various regions' economic dispatch problem was covered, and it was then integrated with the real-world scenario of generators experiencing power limitations. Ultimately, the algorithm is suggested and a flowchart is supplied.

### A. Multi-Region Division

Firstly, we need to express the power as incremental costs. Based on (4), denoting $P$ with $\lambda$ to obtain

$$P_i = \frac{\lambda_i - \beta_i}{2\alpha_i} \tag{5}$$

then substituting (5) into (3), we have

$$\sum_{i=1}^{N} \frac{\lambda_i - \beta_i}{2\alpha_i} = P_D \tag{6}$$

and the optimal power output of node $i$ is

$$P_i^* = \frac{\lambda^* - \beta_i}{2\alpha_i} \tag{7}$$

To solve the ED problem in (3), the power generations are divided into different locations and the distributed method is applied. Administrative, physical, or data distribution meanings might serve as the basis for partitioning recommendations. The index set $1, 2, \ldots, N$ represents the $N$ power generators. Without sacrificing generality, the electricity generation is divided into $M$ zones,

$$\sum_{j=1}^{M} e_j^{\top} \mathbf{P} = P_D \tag{8}$$

where the generator that is a component of the $j$th partition region is either set to zero or to one at the appropriate location of the index vector $e_j$. The ED problem in (3) can thus be expressed in the similar form shown below:

$$\min_{P_i} \sum_{i=1}^{N} f_i(P_i)$$
$$\text{s.t.} \sum_{j=1}^{M} (e_j^{\top} \mathbf{P} - P_{D_j}) = 0 \tag{9}$$
$$P_{\min} \leq P_i \leq P_{\max}$$

where $\sum_{j=1}^{M} P_{D_j} = P_D$ and $\mathbf{P} = (P_1, P_2, \ldots, P_N)^{\top}$.

The ED problem (3) can be solved by the Lagrange multiplier method, which is written as

$$e_j^{\top} \mathbf{h}^{\top} = P_{D_j} \tag{10}$$

where

$$\mathbf{h}^{\top} = (\frac{\lambda_1 - \beta_1}{2\alpha_1}, \frac{\lambda_2 - \beta_2}{2\alpha_2}, \ldots, \frac{\lambda_N - \beta_N}{2\alpha_N}) \tag{11}$$

then, we can get the optimal incremental cost of the power generations

$$\lambda^* = \frac{P_D + \sum_{j=1}^{M} e_j^{\top} \mathbf{B}}{\sum_{j=1}^{M} e_j^{\top} \mathbf{D}} \tag{12}$$

where $\mathbf{B} = (\frac{\beta_1}{2\alpha_1}, \frac{\beta_2}{2\alpha_2}, \ldots, \frac{\beta_N}{2\alpha_N})^{\top}$ and $\mathbf{D} = (\frac{1}{2\alpha_1}, \frac{1}{2\alpha_2}, \ldots, \frac{1}{2\alpha_N})^{\top}$.

### B. Finite-Step Consensus Algorithm Implementation

A distributed finite step consensus technique is presented in this section and applied to the distributed ED issue in various regions[24]. We have improved the formula in Lemma 1, even though the algorithm in this study is substantially similar to the approach in [24]. The $y_i$ updating procedure can be completed by

$$y_i^{(k+1)} = r_{ii} y_i^{(k)} + \sum_{j \in N_i} r_{ij} x_j^{(k)} \tag{13}$$

where the note $x_i$ denotes the graph's topology's $i$th node's state. Furthermore, $y_i$ implies that the parameters must be iterated in accordance with the corresponding $r_{ij}$, where $r_{ij}$ is the row $i$, column $j$, of the created Laplacian matrix in question.

*Lemma 1:* [25] Suppose there is a topology graph that has $D + 1$ different eigenvalues $(\delta_1 \neq \delta_2 \neq \ldots \neq \delta_D + 1)$ in the matrix $L_1$ of its iteration matrix set, where $L_1$ is a Laplace matrix based on the communication topology diagram. And $L_2$ is the constructed Laplace matrix between the divided regions matrix set. Then there are formulas.

$$R^{(k)} = (a^{(k)} + N_{\max} b)\mathbf{I} - bL_2, (k = 1, 2, \ldots, D, b \neq 0) \tag{14}$$

with

$$b = \frac{1}{\prod_{i=2}^{D+1} \sqrt[D]{\delta_i}} \tag{15}$$

$$a^{(k)} = \frac{\delta_{k+1} - N_{\max}}{\prod\limits_{i=2}^{D+1} \sqrt[D]{\delta_i}} \qquad (16)$$

where $N_{\max} = \max\{N_1, N_2, \ldots, N_n\}$, $\mathbf{I}$ is an identity matrix, then we can obtain the consensus in $D$ steps based on (14).

In real systems, load power is scattered throughout rather than existing centrally. Thus, it is rearranged as follows in a multi-agent network topology:

$$\sum_{j=1}^{M} e_j^\top \mathbf{h}^\top = P_D \qquad (17)$$

therefore, the optimal incremental cost is recalculated as

$$\lambda^* = \frac{\sum\limits_{j=1}^{M}(P_{D_j} + e_j^\top \mathbf{B})}{\sum\limits_{j=1}^{M} e_j^\top \mathbf{D}} \qquad (18)$$

Owing to the existence of nodes that surpass power limitations, they are split into two sections: one section does not surpass the limitations, while the other section does. If a set of nodes whose power exceeds the power limits is represented by $\Omega$, then (10) may be rewritten as

$$\overline{\lambda}^* = 2\alpha_i \overline{P}_i + \beta_i, i \notin \Omega \qquad (19)$$

$$\overline{P}_i = \frac{\overline{\lambda}^* - \beta_i}{2\alpha_i}, i \notin \Omega \qquad (20)$$

The power supply-demand balance condition allows us to obtain (21)-(24) by changing (19) to (20)

$$P_D = \sum_{i \in \Omega} \overline{P}_i + \sum_{i \notin \Omega} \overline{P}_i \qquad (21)$$

$$P_j = e_{(1,j)}^\top \mathbf{P} + e_{(2,j)}^\top \mathbf{P} \qquad (22)$$

$$P_D = \sum_{i \in \Omega} e_{(1,j)}^\top \mathbf{G} + \sum_{i \notin \Omega} e_{(2,j)}^\top \mathbf{G} \qquad (23)$$

$$\lambda^* = \frac{(P_D - \sum\limits_{i \in \Omega, j=1}^{M} e_{(1,j)}^\top \mathbf{P}) + \sum\limits_{i \notin \Omega, j=1}^{M} e_{(2,j)}^\top \mathbf{B}}{\sum\limits_{i \notin \Omega, j=1}^{M} e_{(2,j)}^\top \mathbf{D}} \qquad (24)$$

where $\mathbf{G} = (g_1(\overline{P}_1), g_2(\overline{P}_2), \ldots, g_N(\overline{P}_N))^\top$, the vector $e_{(1,j)}^\top$ corresponding to nodes in region $j$ that do not exceed power constraints and $e_{(2,j)}^\top$ corresponds to nodes in region $j$ that do not surpass power constraints, whereas the vector $e_{(2,j)}^\top$ corresponds to nodes in region $j$ that do surpass power constraints. In the distributed multi-region system considered in this research, the optimal value is obtained when the incremental costs within and between areas are consistent. Therefore, the study's algorithm merely needs to figure out what this incremental cost's optimum value is. This algorithmic incremental cost has a value that is applicable to all areas.

The distributed finite-step iteration algorithm presented in this research can be represented by method 1 and Figure 1, taking into account the power output restrictions of all generators.

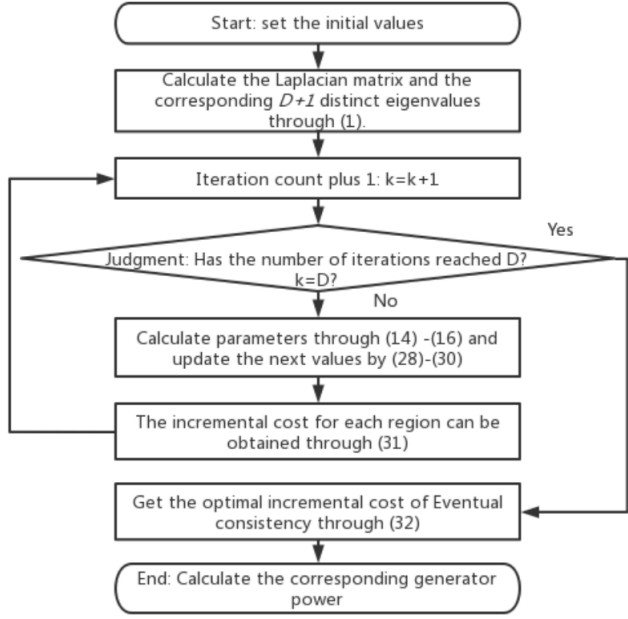

Fig. 1: Flowchart of the distributed finite-step consensus algorithm

## IV. CASE STUDIES

This section includes three case studies that illustrate the efficacy of the suggested distributed control approach. The first case study demonstrates the iterative algorithm's accuracy in two scenarios: one without constraints and the other with constraints. The second case study validates the iterative approach's accuracy under a range of region partitioning modalities. A generator that is dependent on the second case is dynamically replaced by the third case. The connection topology of the micro-grid system, an IEEE 30-bus system with six generators[26], is depicted in Figure 2[27]. Table I lists each node's pertinent parameter. The power curves of these two generators will overlap in the case simulation that follows since parameters 5 and 6 are designed identically to simulate the real-world scenario of having the same motor.

TABLE I: Parameters for 6 nodes network topology[27]

| Nodes | $\alpha_i$ | $\beta_i$ | $\gamma_i$ | $(P_i^{\min}, P_i^{\max})$/kW |
|-------|-----------|-----------|-----------|-------------------------------|
| 1 | 0.00164 | 7.75 | 420 | (100,260) |
| 2 | 0.00215 | 7.80 | 208 | (150,400) |
| 3 | 0.00395 | 7.62 | 172 | (150,300) |
| 4 | 0.00172 | 7.84 | 352 | (100,400) |
| 5 | 0.00368 | 7.73 | 178 | (80,320) |
| 6 | 0.00368 | 7.73 | 178 | (80,320) |

**Algorithm 1** The finite-step iteration algorithm with constraints

1: From Lemma 1, $b$ and $a^0$ can be calculated. And set the initial value $W_j^0, Y_j^0$ and $Z_j^0$, where $W_j^0, Y_j^0$ and $Z_j^0$ represents portions of the numerator and denominator in (24):

$$W_j^0 = P_j - \overline{P}_c, c \in \Omega \cap i \in R_j \tag{25}$$

$$Y_j^0 = \sum_{j=1}^{M} \overline{e}_j^\top \mathbf{B}, i \notin \Omega \tag{26}$$

$$Z_j^0 = \sum_{j=1}^{M} \overline{e}_j^\top \mathbf{D}, i \notin \Omega \tag{27}$$

2: Using (1), determine the Laplacian matrix and the associated $D+1$ unique eigenvalues. Moreover, compute $b$ and $\alpha^{(k)}$ via (15) and (16), obtaining $R^{(k)}$ via (14);

3: The following iterations, where $r_{jj}$ and $r_{jq}$ denote the corresponding member in the $R^k$ matrix, update the subsequent values:

$$W_j^{(k+1)} = r_{jj} W_j^{(k)} + \sum_{q \in N_i} r_{jq} W_q^{(k)} \tag{28}$$

$$Y_j^{(k+1)} = r_{jj} Y_j^{(k)} + \sum_{q \in N_i} r_{jq} Y_q^{(k)} \tag{29}$$

$$Z_j^{(k+1)} = r_{jj} Z_j^{(k)} + \sum_{q \in N_i} r_{jq} Z_q^{(k)} \tag{30}$$

4: Calculate the incremental costs of region j at each iteration:

$$\overline{\lambda}_j^k = \frac{W_j^{(k)} + Y_j^{(k)}}{Z_j^{(k)}} \tag{31}$$

5: Following $D$ iterations, the ideal incremental cost can be determined as follows:

$$\overline{\lambda}^* = \frac{1}{M} \sum_{j=1}^{M} \overline{\lambda}_j^{(k)} \tag{32}$$

6: Next, determine node $i$'s power output for each iteration. The power $P_i^*$ is equal to the corresponding upper and lower boundaries if $P_i^*$ exceeds the upper and lower bounds of the power constraint:

$$P_i^* = \begin{cases} \frac{\overline{\lambda}^* - \beta_i}{2\alpha_i}, & i \notin \Omega \\ P_i^{\min} \text{or} P_i^{\max}, & i \in \Omega \end{cases} \tag{33}$$

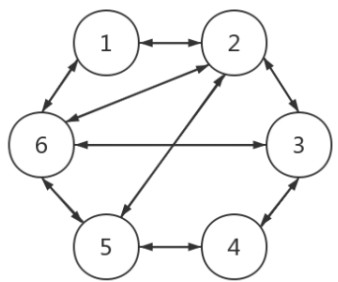

Fig. 2: Topology diagram of distributed connection for six generators[28]

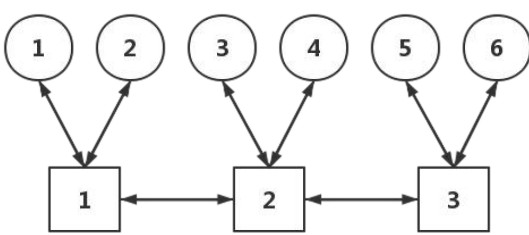

Fig. 3: The diagram of six generators divided into three areas[28]

### A. Case1: comparison between algorithms with and without power constrained

To validate the algorithm described in this research, Case 1 primarily splits the IEEE 30-bus system with 6 generators into 3 zones with $P_D = 1200$kW, as illustrated in Figure 3. The example confirms that the finite-step iterative approach suggested in this work is accurate for both constrained and unconstrained power scenarios.

We start by ignoring each node's power limitation. Without limitations, we are able to determine $e_1^\top = (1,1,0,0,0,0)^\top$, $e_2^\top = (0,0,1,1,0,0)^\top$ and $e_3^\top = (0,0,0,0,1,1)^\top$. Based on the network topology, the Laplacian matrix $L_1$ and $L_2$ are expressed as

$$L_1 = \begin{bmatrix} 2 & -1 & 0 & 0 & 0 & -1 \\ -1 & 3 & -1 & 0 & -1 & -1 \\ 0 & -1 & 2 & -1 & 0 & -1 \\ 0 & 0 & -1 & 2 & -1 & 0 \\ 0 & -1 & 0 & -1 & 3 & -1 \\ -1 & -1 & -1 & 0 & -1 & 4 \end{bmatrix}$$

and

$$L_2 = \begin{bmatrix} 1 & -1 & 0 \\ -1 & 2 & -1 \\ 0 & -1 & 1 \end{bmatrix}$$

Since $L_1$ has six distinct eigenvalues, $D = 5$ may be computed, indicating that consistent results can be obtained by repeating five stages.

The recommended method can be applied without limitations to obtain the results shown in Figures 4(a) and 5(a). All

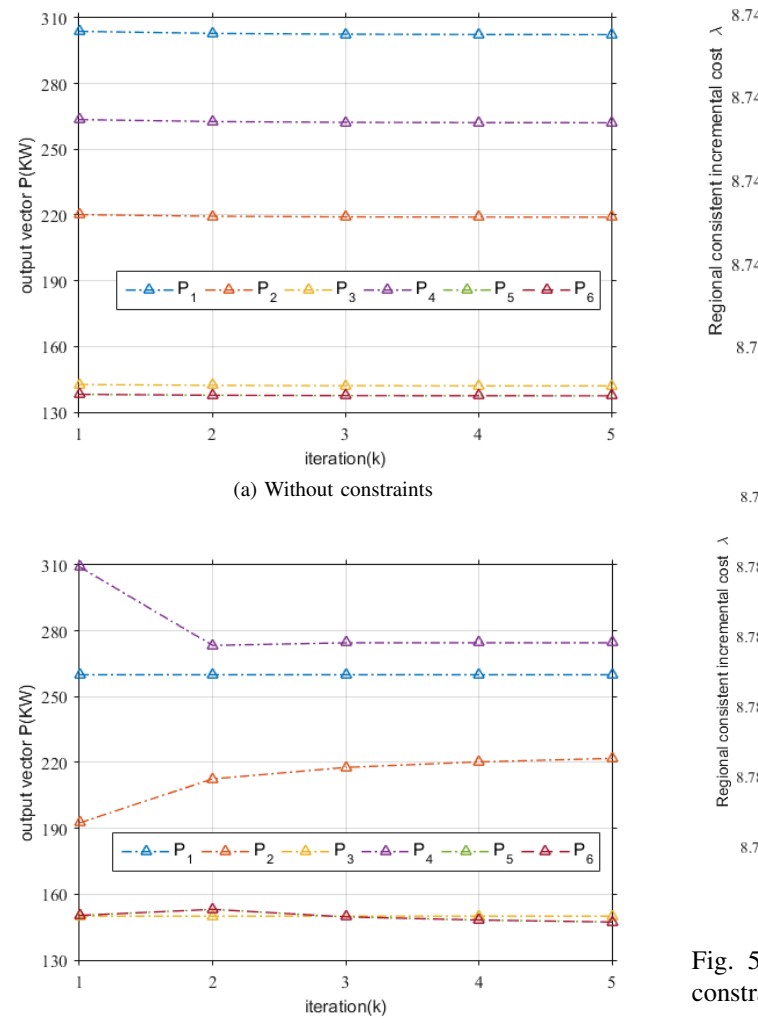

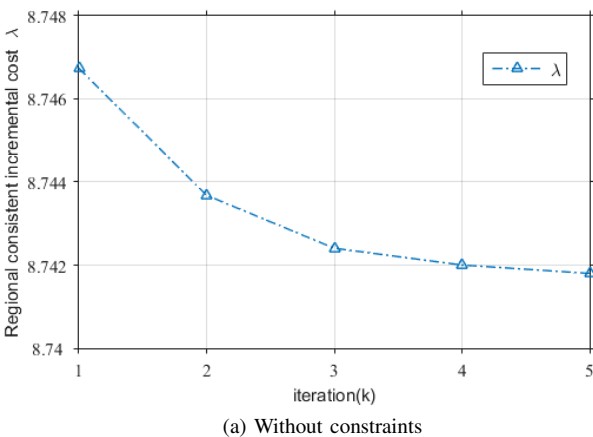

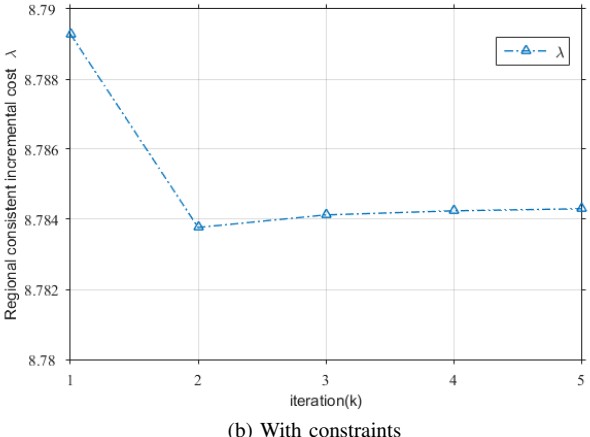

(a) Without constraints

(b) With constraints

Fig. 4: The iterative results of generator power with both constrained and unconstrained power

(a) Without constraints

(b) With constraints

Fig. 5: The iterative results of incremental cost with both constrained and unconstrained power

nodes' power updates are displayed in Figure 4(a), totaling 1200kW. Moreover, the incremental cost iteration update is shown in Figure 5(a). We can thus acquire each power in Figure 4(a) for the following values: $P_1 = 302.3780$kW, $P_2 = 219.0232$kW, $P_3 = 142.0000$kW, $P_4 = 262.1512$kW, $P_5 = 137.4728$kW, and $P_6 = 137.4728$kW. The total incremental cost in Figure 5(a) is 8.7418. In the example case, the generator is running freely. It makes it clear that generators 2 and 3 have exceeded their permitted boundaries.

Thus, under constraints, $e_{(1,1)} = (1,0,0,0,0,0)^\top$, $e_{(1,2)} = (0,0,1,0,0,0)^\top$, $e_{(1,3)} = (0,0,0,0,0,0)^\top$, $e_{(2,1)} = (0,1,0,0,0,0)^\top$, $e_{(2,2)} = (0,0,0,1,0,0)^\top$, $e_{(2,3)} = (0,0,0,0,1,1)^\top$ can be obtained. The simulation results can be achieved by applying the suggested algorithm, which is provided in Figures 4(b) and 5(b), while taking each node's power constraint into consideration. 8.7843 is the total incremental cost shown in Figure 5(b). Additionally, $P_1 = 260.0000$kW, $P_2 = 228.7828$kW, $P_3 = 150.0000$kW, $P_4 = 274.4887$kW,

$P_5 = 143.2472$kW, and $P_6 = 143.2472$kW are the best values for each node in Figure 4(b). Figure 4(b) illustrates how each node's power updates over time, eventually converges over five steps, and totals 1200kW. Additionally, the generator power output is constrained within the lower and upper limitations at each iteration step. Furthermore, Case 1's simulation results demonstrated consistency in just four stages, confirming the algorithm's validity.

### B. Case2: change of the total power generated

This case represents the scenario when the system's overall power requirement varies at a specific point in time and is primarily based on Case 1 with limitations. This paper's algorithm's fast finite step iteration speed allows it to respond with accuracy and quickness.

Figure 6 illustrates how the total power demand shifts from 1200kW to 1500kW at time $t_1$. To attain consistency in a brief amount of time, the algorithm iterates once more. Using the finite step technique, we can determine that the final incremental cost under a total power requirement of 1500kW is 9.11. Additionally, $P_1 = 260.0000$kW, $P_2 = 305.2769$kW, $P_3 = 188.9482$kW, $P_4 = 369.9683$kW, $P_5 = $

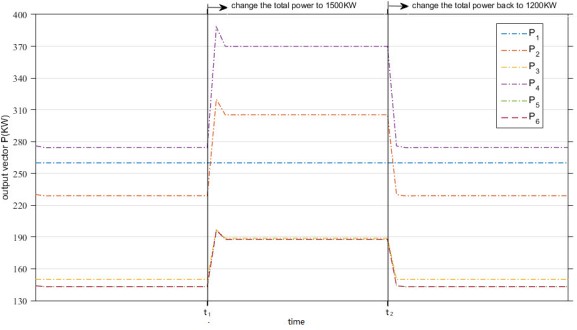

Fig. 6: The results of changing the total power generated at time $t_1$ and $t_2$

187.8656kW, and $P_6 = 187.8656$kW are the best values for each node.The total power returns to 1200kW at time $t_2$ from 1500kW. Additionally, the incremental cost rises to 8.7843. Furthermore, $P_1 = 260.0000$kW, $P_2 = 228.7828$kW, $P_3 = 150.0000$kW, $P_4 = 274.4887$kW, $P_5 = 143.2472$kW, and $P_6 = 143.2472$kW are the ideal values for each generator.

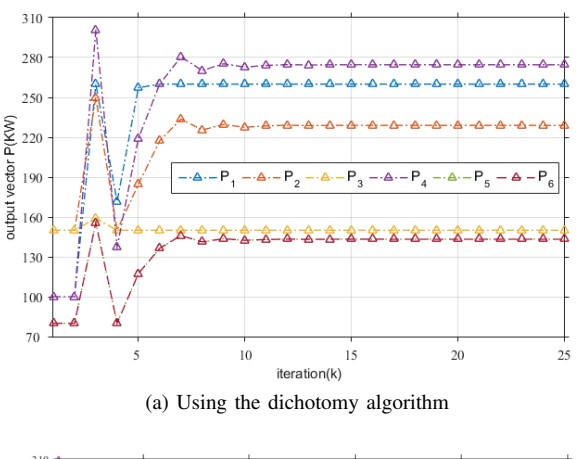

(a) Using the dichotomy algorithm

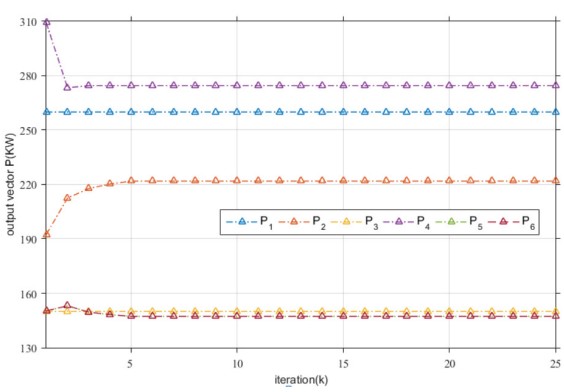

(b) Using the distributed finite-step consensus algorithm

Fig. 7: Comparison of power iteration results between two algorithms

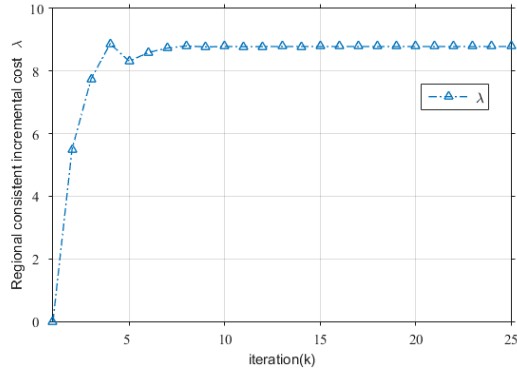

(a) Using the dichotomy algorithm

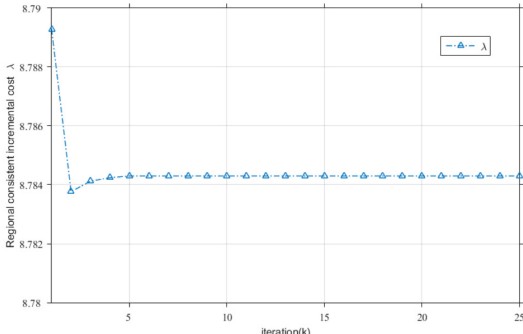

(b) Using the distributed finite-step consensus algorithm

Fig. 8: Comparison of consistent incremental cost iteration results between two algorithms

*C. Case3: comparison between the proposed algorithm and the general dichotomy method*

The classic generic dichotomy algorithm, whose parameters are the same as those with limitations in Case 1 above, is compared with the approach in this paper in this Case Study. These are Figures 7(a) and 8(a), which we can acquire by applying the dichotomy search approach. Additionally, Figures 7(b) and 8(b) exhibit the constrained outcomes from Case 1 above.

The suggested approach needs less iterations to reach the same result, whereas the dichotomy method requires more convergence time, as can be shown by comparing the iterative results of the generating power of the two ways in Figure 7. Subsequently, Figure 8's comparison of the iterative outcomes of the incremental costs of the two techniques demonstrates that, in contrast to the suggested method, the dichotomous finding method needs more than 20 iteration steps to attain convergence. The finite step method outperforms this technique in terms of iteration times, as more time may be saved with fewer iterations. This suggests that there are more benefits and more.

## V. CONCLUSION

This study proposes a hybrid centralized-decentralized ED technique with divided regional incremental costs, based on

a distributed finite-step consensus process. This approach genuinely considers the generator's power limitations and reaches regionally consistent convergence in a few number of iterative steps. Initially, in a multi-agent power system, a Laplacian matrix can be derived from the distributed topology of the generator. Subsequently, a constructed Laplace matrix that illustrates the communication link between geographically split zones can be obtained. After the final additional cost achieves the regional consistency, we compute the incremental cost of each region while accounting for the generators' power limitations. In addition, during the iteration process, the output power of the distributed generator can be limited to the corresponding range, which can ensure the balance between power supply and demand. Finally, the correctness, rationality and speed of the finite-step iteration algorithm are verified through three calculation examples.

## ACKNOWLEDGMENTS

This work was supported in part by the National Natural Science Foundation of China under Grant 62203088, 51939001, 61976033, 62273072.

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
