# OpenReview forum: "Hybrid Centralized-Decentralized Economic Dispatch Based on A Distributed Finite-Step Consensus Algorithm with Divided Regional Incremental Costs"
_IEEE.org/ICIST/2024/Conference — IEEE ICIST 2024 Conference Submission_

### Official Review · Reviewer_3dK3 · 2024-08-21
**This article is quite fascinating and of high quality.**

**Rating:** 7
**Confidence:** 3

**Review:**

The paper titled "Hybrid Centralized-Decentralized Economic Dispatch Based on A Distributed Finite-Step Consensus Algorithm with Divided Regional Incremental Costs" proposes a distributed finite-step consensus method that incorporates geographical region division to achieve economical power allocation without the need for full graph connectivity. Firstly, authors derive a Laplacian matrix from the distributed topology of generators in a multi-agent power system. Subsequently, authors construct another Laplacian matrix based on geographical partitioning to illustrate the communication relationships between these regions. Then, taking into account the power constraints of the generators, we calculate the incremental cost for each region. Finally, the comparison experiments are proposed to illustrate the effectiveness of the proposed method. My specific feedback is as follows: 1) The research challenges and innovations of this paper are not
explained in this paper. Please add these details.2) Some formatting issues need to be addressed.

---

### Official Review · Reviewer_cQrp · 2024-08-22
**This article is very interesting and a good one**

**Rating:** 7
**Confidence:** 3

**Review:**

This paper delved into a distributed finite-step consensus approach that integrates geographical region division, with the objective of achieving cost-effective power allocation without relying on full graph connectivity. The obtained result is valuable and can be accepted if the following problems can be clarified.
(1) In the introduction, the shortages of those relevant studies are suggested to be further summarized.
(2) In the end of Section 1, the organization of this study is suggested to be summarized.
(3) The article uses "Figure" and the caption of the image uses "fig", which needs to be unified between the two in the simulation section.
 (4) In the simulation section, more analysis can be added to better explain the main results of this paper, that's not enough.
 (5) The future work is missing in the Conclusion.
 (6) There exist several spelling and grammar errors. Please check carefully and further polish

---

### Official Review · Reviewer_i9Bw · 2024-08-27
**In this paper, the author proposes a hybrid centralized-decentralized ED technique with divided regional incremental costs, based on distributed finite-step consensus process.**

**Rating:** 7
**Confidence:** 3

**Review:**

a The abstract should mainly include elements such as research purpose, methods and final results, and reviewers suggest optimizing the content of the abstract.
b What are the significant differences between this study and previous studies? The author needs more explicit emphasis.
c There are some grammatical mistakes and typos. Please examine the full text further and revise them.

---

### Comment · Reviewer_cQrp · 2024-08-21
**This article is very interesting and a good one**

This paper delved into a distributed finite-step consensus approach that integrates geographical region division, with the objective of achieving cost-effective power allocation without relying on full graph connectivity. The obtained result is valuable and can be accepted if the following problems can be clarified.

(1)	In the introduction, the shortages of those relevant studies are suggested to be further summarized.
(2)	In the end of Section 1, the organization of this study is suggested to be summarized.
(3)	The article uses "Figure" and the caption of the image uses "fig", which needs to be unified between the two in the simulation section.
(4)	In the simulation section, more analysis can be added to better explain the main results of this paper, that's not enough.
(5)	The future work is missing in the Conclusion.
(6)	There exist several spelling and grammar errors. Please check carefully and further polish

---

### Decision · Program_Chairs · 2024-09-06

Accept (Oral)